# Updating the remembered position of targets following passive lateral translation

**John J. J. Kim**  *, **Laurence R. Harris**

Department of Psychology, York University, Toronto, Ontario, Canada

* johnk84@yorku.ca

## Abstract

Spatial updating, the ability to track the egocentric position of surrounding objects during self-motion, is fundamental to navigating around the world. However, people make systematic errors when updating the position of objects after linear self-motion. To determine the source of these errors, we measured errors in remembered target position with or without passive lateral translations. Self-motion was presented both visually (simulated in virtual reality) and physically (on a 6-DOF motion platform). People underestimated targets' eccentricity in general even when just asked to remember them for a few seconds (5–7 seconds), with larger underestimations of more eccentric targets. We hypothesized that updating errors would depend on target eccentricity, which was manifested as errors depending not only on target eccentricity but also the observer's movement range. When updating the position of targets within the range of movement (such that their actual locations crossed the viewer's midline), people overestimated their change in position relative to their head/body compared to when judging the location of objects that were outside the range of movement and therefore did not cross the midline. We interpret these results as revealing changes in the efficacy of spatial updating depending on participant's perception of self-motion and the perceptual consequences for targets represented initially in one half of the visual field having to be reconstructed in the opposite hemifield.

## Introduction

Most people can easily navigate around their environment and effectively keep track of their surroundings. To do so they must update the perceived position of surrounding objects relative to themselves as they move. This process is called spatial updating [1–3]. Spatial updating is strongly influenced by the observer's perception of the spatial structure of their environment, i.e., the egocentric directions and distances of the objects that make up the scene [4, 5], and by their self-motion [4]. Knowing about their own movement requires integrating sensory information from visual (optic flow), vestibular, somatosensory and motor systems [6–8], as well as incorporating internal knowledge of their planned movement, during an active movement. Combining motion cues from these sources allows a person to know about their own movement and hence, theoretically, to predict the new positions of surrounding earth-stationary objects relative to them after their movement [8].

**Data Availability Statement:** All data files are available from the OSF database (https://osf.io/mb8ga/?view_only=804d889fd32d4a9bb395b01f9f4cefaa).

**Funding:** These experiments were supported by a grant from Natural Sciences and Engineering

Research Council of Canada RGPIN-2020-06093 to
LRH. There was no additional external funding
received for this study.

**Competing interests:** The authors have declared
that no competing interests exist.

Any misperception of their travel distance will lead to errors in computing the new egocentric locations of objects after a move. In real life, such errors can usually be corrected by visual information from the environment that allows an observer to derive object locations using allocentric references, i.e., landmarks [4]. To prevent people using such landmarks to derive object locations after a move, as well as to segregate the effect of vestibular feedback from the visual feedback on people's spatial updating, previous studies have usually deprived their participants of visual information while moving by conducting the movement in complete darkness [9–11]. However, the visual motion signal (i.e., optic flow) provides important information about both a person's movement and the relative movement of the surrounding environment as a person move through it [12].

How the position of an object changes relative to an observer is substantially different depending on whether movement is rotational or linear: updating an object's position during linear movement requires more complex computation because only linear movement alters the distance between the observer's body and the object, i.e., the egocentric distance. During rotation, the egocentric distance of objects remains constant. Therefore, the only updating required during rotation is to displace everything by the angle through which the person rotates. During linear movement, however, both the egocentric distance and direction of all the objects in a scene change differently. For example, the egocentric direction of an object will not change at all as a person moves directly towards or away from it. In this special case, the only updating required is to the egocentric distance of the object. When a person updates the position of an object in front of them while moving laterally, both direction and distance to the object change. Therefore, updating errors made during the movement may be due to mis-updating the direction or distance to the target, or both.

There is no shortage of literature looking into spatial updating as it is a fundamental and necessary ability for people's everyday lives. Most of these studies look at spatial updating following rotational movements, specifically yaw rotations [3, 5, 13] and sometimes forward translation [14, 15] but there are few studies considering the consequences of linear lateral movement. Klier et al. [11] looked at updating after passive linear translation (fore-aft, up-down and left-right) but only used targets directly in front of the observer. Here, we conducted three experiments using lateral translation and targets at various eccentricities to compare updating performance for objects that start and end at different eccentricities.

## Hypotheses

In experiment 1 (Exp 1), participants updated a target's position in virtual reality (VR) with and without visually simulated lateral translation. Comparing the errors following translation with the errors without translation tells us the error associated with updating a target's position. Gutteling and Medendorp [9, 11] showed that the remembered position of a target after a passive lateral translation in dark was associated with errors in the direction of movement. We hypothesized that we would find similar movement direction dependent errors. People already underestimate target eccentricity without any self-motion [16, 17] even when tested after just a few moments of darkness. In one study participants undershot when they reached for target LEDs with targets from 0.2 to 0.4 m on their right using their right hand [16]. Another study had participants point at targets at different eccentricities (10–40˚) arranged at a constant distance. Participants generally underestimated target eccentricity as well as distance [17]. In both studies, the underestimation was greater for more eccentric targets. After a lateral translation, the eccentricity of a target changes and so the amount of underestimation may also change. Therefore, we hypothesized that updating errors would differ between targets of different eccentricities.

Experiment 1 was conducted in VR and therefore participants were only exposed to visual motion. However, physical motion cues (e.g., gravito-inertial-somatosensory cues) play an important role in self-motion perception [7]. In experiment 2 (Exp 2), we varied physical and visual motions cues using VR and a motion platform to test whether more self-motion cues would reduce updating errors. Our hypothesis was that people would be more accurate at updating target positions after being translated laterally when physical motion cues were available, i.e., when self-motion perception was more accurate.

Due to the physical limitation of the motion platform used in Exp 2, the translation distance had to be shorter than the distances used in Exp 1. Gutteling and Medendorp demonstrated that the magnitude of updating errors depends on travel distance [9], but the longest travel distance they used was 0.18 m (over 3 seconds) which is much shorter than the 0.46 m (over 5 seconds) used in our experiments. To evaluate the effect of travel distance in updating, specifically to compare the different distances used in Exp1 (1 m) and Exp 2 (0.46 m), we varied the translation distance in experiment 3 (Exp 3)—short vs. long distance. We hypothesized that updating errors would be larger with long travel distance compared with short travel.

## Materials and methods

### Overview

Here we report three experiments we conducted investigating updating errors after lateral translation. A target ball was presented in VR at various eccentricities. The target then disappeared, and the participant was moved to the left or right. They then had to report the remembered location of the target ball.

### Participants

Exp 1: twenty participants (14 females and 6 males, average age of 20.55 years; SD = 3.72) were recruited between 11/28/2018 and 12/10/2018, Exp 2: twenty-four participants (14 females and 10 males, average age of 20.75 years, SD = 3.98) were recruited between 01/29/2019 and 03/12/2019, and Exp 3: Twenty-six participants (20 females and 6 males, average age of 24.20 years, SD = 11.69) were recruited between 02/28/2020 and 06/02/2022. All participants were undergraduate students at York University recruited via Undergraduate Research Participant Pool (URPP) participated in the study. They signed a written informed consent form and were given course credit for their participation. The experiments were approved by the York University's Ethics Review Board. All participants had normal or corrected-to-normal vision and were right-handed.

### Apparatus

All tests were conducted in VR using an Oculus Rift Head Mounted Display (HMD). The HMD was an Oculus Rift Consumer Version 1 with a resolution of 1080×1200 pixels per eye, maximum 90 Hz refresh rate, a 110˚ field of view, and weighed 470 grams. In Exps 1 and 2, participants also used an Oculus Touch controller in their right hand to point at target positions. In Exp 3, participants used a keyboard. The HMD was powered from a Windows 10 computer: Exps 1 and 2 –model Alienware 17 R4 with Intel® Core™ i7-7700HQ CPU @ 2.80 GHz, 16.0 gigabyte RAM, and NVIDIA GeForce GTX 1070 graphics card, Exp 3 –Alienware Area-51 R2 equipped with Intel® Core™ i7-5820K Processor, 16.0 gigabyte RAM, and a NVIDIA GeForce GTX 980 graphics card. The VR environment was built in Unity Game Engine (version 2018.3.14f1) and the test was programmed with C# language.

In Exp 2, the MOOG Motion System (model 6DOF2000E-170E122A), a six-degree of freedom motion base platform, was used to produce physical motion. We restrained participants' neck movement using a neck brace (4098 OPPO Cervical Collar) to prevent them from tilting or injuring their neck during movement. The MOOG was controlled from a Windows 10 computer, model Dell with Intel® Core™ i7-2600 CPU @ 3.40 GHz, 8.0 gigabyte RAM, and AMD Radeon HD 5450 graphics card. The motion profiles were programmed in Unity game engine with C# language.

## Stimuli

The test environment in VR was a detailed virtual rendition of the actual room in which the MOOG motion platform resides at the Sherman Health Science Research Centre at York University. The virtual room provided a rich environment including a simulated screen with a projector, a seat attached to the MOOG, and an avatar body. Fig 1 shows the real MOOG room (B) and the virtual rendition of the room (C). An image of a tennis ball, diameter 67 mm, was used for the updating test and was presented on a simulated screen. Target locations were spread out horizontally from the center of the screen at eye height. The targets were shown one at a time for each trial. The target locations for each experiment are detailed below.

Exp 1: The experiment took place in an office-like test room (Fig 1A) with participants wearing an HMD in which the virtual test room was simulated (Fig 1C). The potential targets were positioned at between -0.75 m to +0.75 m from the center of the screen in 0.25 m increments (7 total), and were displayed one at a time, rendered onto the virtual screen that formed part of the simulation (1.5 meters x 2 meters) 2 m away from the viewing camera. There were two initial observer positions: -0.5 m (Left) and +0.5 m (Right). In the Stationary condition, observer stayed at the initial position throughout the run. In the Visual Motion condition, participants experienced one meter of visually simulated passive–i.e., without them taking any voluntary action–lateral movement (by moving the camera through the virtual environment) from one observer start position to the other (1 m travel), left-to-right, or right-to-left. The movement profile was a linear lateral motion at a constant velocity of 0.143 m/s for 7 seconds then rapidly decelerated to a stop. During the lateral translation, randomly generated dots were displayed on the screen to provide more optic flow during the observer's simulated translation. The density of the dots was 50 dots per m$^2$. The size of the dots varied between 1.0 cm and 6.5 cm. Exp 1 consisted of 112 trials in total (2 motion conditions × 2 observer positions × 7 target positions × 4 trials per target).

Exp 2: The experiment took place in a MOOG test room (Fig 1B) with participants wearing an HMD showing the identical virtual test room (Fig 1C). The translation distance had to be shortened compared to Exp 1 because of the physical limitation of the MOOG motion platform and the target positions were also changed accordingly with a correspondingly smaller range. The potential targets were positioned at between -0.46m to +0.46m from the center of the screen in 0.23 m increments (5 total), displayed one at a time, on the virtual screen (1.5 meters x 2 meters) 1.75 m away from the viewing camera. There were two initial observer positions in front of the targets at -0.23 m (left) and +0.23 m (right). As in Exp 1, the observers stayed at one of these positions in the stationary condition. In the moving conditions (Visual Motion, Physical Motion, or Full Motion–i.e., with both cues–conditions), they were moved passively between these positions, either left-to-right or right-to-left depending on their initial position. For the Full Motion condition, the visual motion information was consistent with the physical head movement of the observer tracked by the HMD. The movement profile was a linear lateral motion at a constant velocity of 0.092 m/s, translating for 5 seconds (total travel = 0.46 m; the maximum lateral travel distance of the MOOG) for both visual and

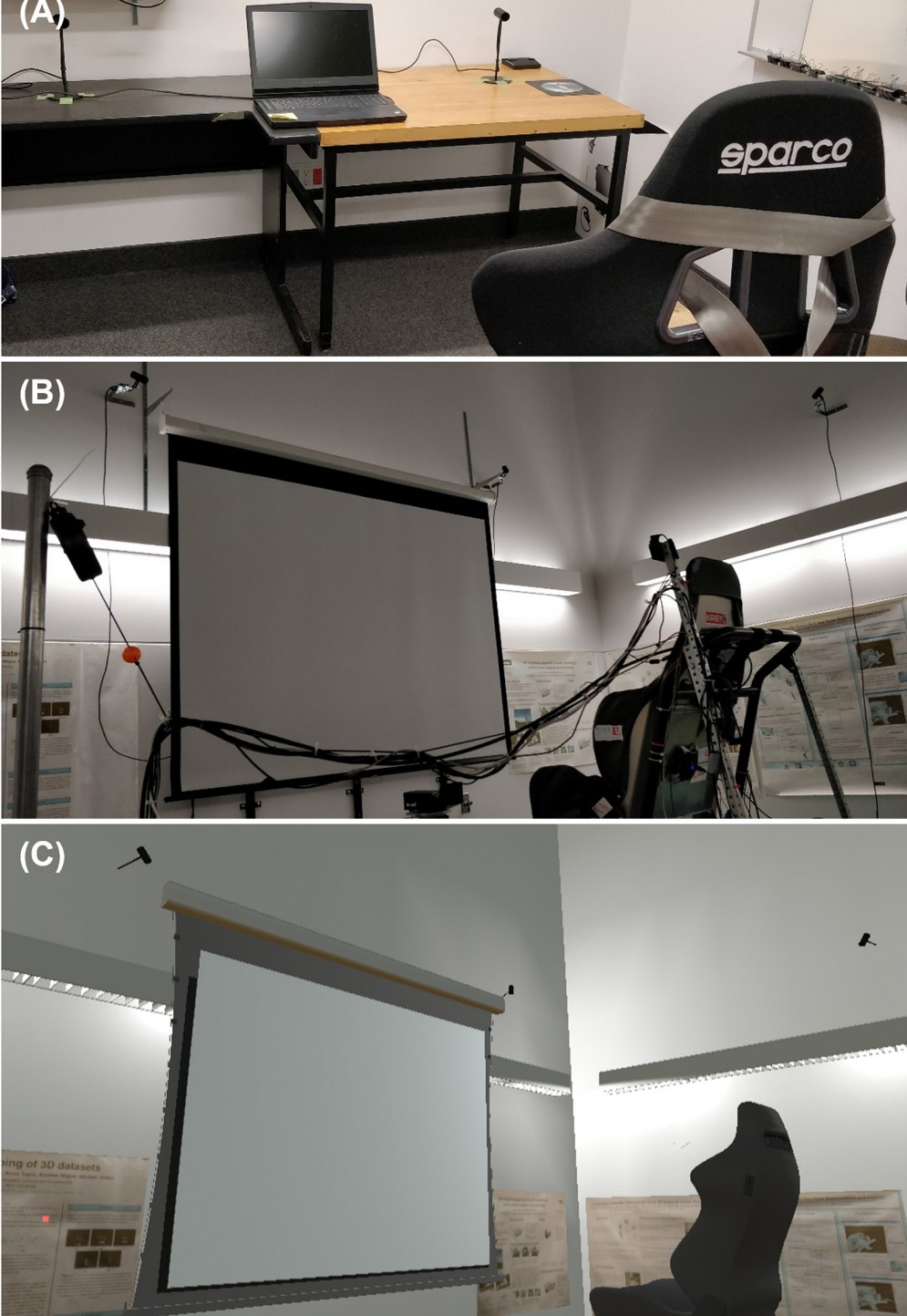

**Fig 1. Experiment setup photos.** (A) Photo of the office-like test room with a chair used for Exp 1 and 3. (B) Photo of the MOOG test room with a seat on the motion platform used to produce physical motion cues. (C) Example image of the virtual rendition of the MOOG test room (shown in B) with a virtual seat and a screen. This virtual rendering of the room was used for all experiments (Exps 1, 2, and 3).

physical motions. For physical motion cues, the MOOG accelerated rapidly to the constant velocity. At the end of travel, the MOOG decelerated rapidly until stop. As in Exp 1, during the lateral translation, randomly generated dots were displayed on the screen except for the Physical motion condition where the HMD only showed empty dark screens (i.e., no visual motion information). Exp 2 consisted of 160 trials in total (4 motion conditions × 2 observer positions × 5 target positions × 4 trials per target).

Exp 3: The experiment took place in an office-like test room (Fig 1A) with participants wearing an HMD in which the virtual test room was simulated (Fig 1C). The targets were between -1.15 m to +1.15 m from the center of the screen in 0.23 m increments (11 total) and were displayed one at a time, on the virtual screen (1.5 meters x 3.5 meters) 1.75 m away from the viewing camera. The observer was always initially positioned in front of the target at 0m (center of the screen). In the visual motion conditions, they were moved passively either to the left or to the right side (visually only) for 5 seconds. Two translation distances were used: 0.46 m for the Short travel condition and 1 m for the Long travel condition. During translations, random dots appeared on the screen which were made to blink between 0.5 to 1 Hz. Exp 3 consisted of 198 trials in total (3 translation distances × 2 motion directions × 11 target positions × 3 trials per target).

During the response phase, a red dot was displayed on a black wall at the same distance as the screen, which participants moved to indicate their remembered target position on the screen. The parameters for each of the experiments are summarized in Table 1.

## Training

Every participant went through a training session at the start of the test session to get familiar with using the controller/keyboard. They were shown the visual target (an actual size image of a tennis ball) on the screen. They had to move a red dot to the position of the target using the controller (Exps 1 and 2) or the left/right arrow keys on the keyboard (Exp 3). Once the red dot was on the target, they pressed a button to "hit" the target. If they pressed the button when the red dot was not on the target, it was considered as they "missed" the target. The training was done for 10 trials.

**Table 1. Experiment parameters.**

| Parameters | Experiments | | |
|---|---|---|---|
| | **Exp 1** | **Exp 2** | **Exp 3** |
| Screen Size (m × m) | 1.5 × 2 | 1.5 × 2 | 1.5 × 3.5 |
| Viewing Distance (m) | 2 | 1.75 | 1.75 |
| Travel Distance (m) | 0 and ±1 | 0 and ±0.46 | 0, ±0.46, and ±1 |
| Travel Time (s) | 7 | 5 | 5 |
| Travel Speed (m/s) | 0.143 | 0.092 | 0.2 and 0.092 |
| Motion Conditions | Stationary, and Vision Motion | Stationary, Vision Motion, Physical Motion, and Full Motion (vision + physical) | Stationary, Short Travel, and Long Travel |
| Observer Start Position (m) | ±0.5 | ±0.23 | 0 |
| Dots on screen during translation | Continuous | Continuous | Blinking at .5–1 Hz |
| Target Positions from the Observer (m) | ±0.75, ±0.5, ±0.25, and 0 | ±0.46, ±0.23, and 0 | ±1.15, ±0.92, ±0.69, ±0.46, ±0.23, and 0 |
| Experimental Conditions | 7 (Targets) × 2 (Motion Conditions) | 5 (Targets) × 4 (Motion Conditions) | 11 (Targets) × 3 (Travel Distances) |

## Procedures

**Experiment 1.** Participants sat on a chair in an office-like test room (Fig 1A). Then they put on the HMD in which they viewed the virtual environment and the stimuli (Fig 1C). Participants held an Oculus Touch controller with their right hand which they used to respond in each trial. They were instructed to look straight at the simulated screen in the virtual environment (Fig 1C). The experiment was done in blocks where each block corresponded to one of the motion conditions: Stationary, and Visual Motion. The block orders were counterbalanced. Trials were sequenced to move participants between the two observer positions (i.e., left-to-right motion followed by the right-to-left motion), but the targets were presented in a randomized order.

For each trial, a bell sound was played to let participants know the trial was starting. A) Then a fixation cross was displayed in the middle of their visual field which they were told to fixate. B) After 0.5 seconds, a visual target (the same image of a tennis ball from the training) appeared for 0.5 seconds at some eccentricity from the observer (see Table 1). C) After the visual target disappeared, randomly generated static dots were shown on the screen. D) During the translation phase, participants were either stationary (Stationary condition) or passively moved laterally to the left or to the right (Visual Motion condition). During the visual motion they moved relative to the screen with the dots–optic flow. The initial starting position was selected randomly (Left or Right) then alternated between the two positions. The movement direction depended on the observer's starting position, and they were then moved to the opposite observer position. E) After the movement, or after an equivalent idle period in the stationary condition, the screen went dark, and a second bell sound was played. When the participants heard the second bell, they placed the red dot at the remembered target location with the controller (the pointing location was indicated on the screen with a red dot as in the training). When the red dot was at the position where the participant thought the target was, they pressed a button on the controller to indicate their choice. The pointing location (red dot) was recorded as indicating the participant's remembered target location. Fig 2 describes the experimental procedure in detail.

**Experiment 2.** Participants sat on a chair attached to a MOOG motion platform (Fig 1B). Then they put on the HMD in which they viewed the virtual environment and the stimuli (Fig 1C). Participants held an Oculus Touch controller with their right hand which they used to respond in each trial. They were instructed to look straight at the simulated screen in the virtual environment (Fig 1C). The experiment was done in blocks where each block corresponded to one of the motion conditions: Stationary, Visual Motion, Physical Motion, and Full Motion. The block orders were counterbalanced. As in Exp 1, trials were sequenced to move participants between the two observer positions (i.e., left-to-right motion followed by the right-to-left motion), but the targets were presented in a randomized order. The procedure followed the same steps as Exp 1 except during the translation phase–D). Fig 2 shows all the step of the procedure in detail. D) When participants were passively moved laterally to the left or to the right (Visual Motion, Physical Motion, and Full Motion conditions), in the Visual and Full Motion conditions they moved relative to the screen with the dots–optic flow. In the Physical Motion condition, the screen was dark during translation–no optic flow. As in Exp 1, the initial starting position was selected randomly (Left or Right) and was then alternated between the two positions. The movement direction depended on the observer's starting position, and they were then moved to the opposite observer position.

**Experiment 3.** Participants sat on a chair in an office-like test room (Fig 1A). Then they put on the HMD in which they viewed the virtual environment and the stimuli (Fig 1C).

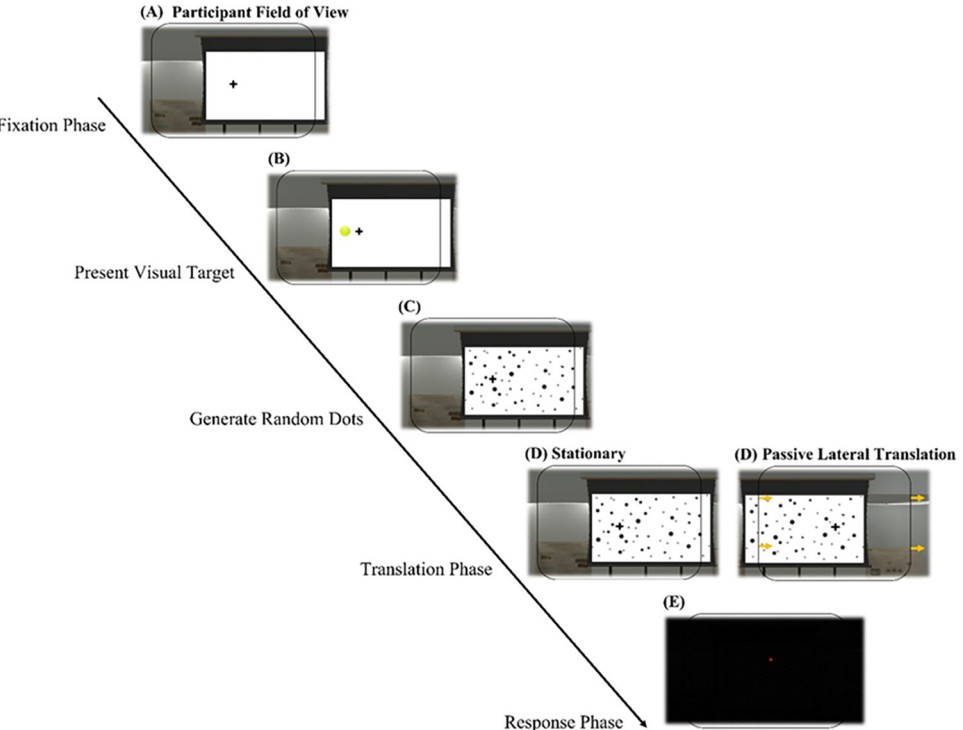

**Fig 2. Experimental procedure.** (A) The participant fixated on the fixation cross for 500 ms. (B) Visual target presented for 500 ms. (C) Randomly generated dots appeared on the screen after the visual target disappeared. (D) The participant was either stationary for a set period or passively moved laterally. (E) The screen turned dark. The participant then moved the indicator target to the remembered location of the visual target as fast as they could.

Participants put their hand on a keyboard which was used instead of the controller used in Exp 1 and 2 to prevent the participants' arms becoming too tired towards the end of the experiment. They were instructed to look straight at the simulated screen in the virtual environment (Fig 1C). The experiment was done in blocks where each block corresponded to one of the motion conditions: Stationary, Short Travel, and Long Travel. The block orders were counterbalanced. The moving directions (in Short and Long Travel conditions) and the target positions were presented in a randomized order within each block.

The procedure followed the same steps as Exp 1 except during the phases D and E. See Fig 2 for each step of the procedure in detail. D) As in Exp 1, when participants were passively moved laterally to the left or to the right (Short Travel and Long Travel conditions) they moved relative to the screen with the dots–optic flow. Instead of starting either on the left or right side of the screen (as in Exp 1 and 2) the participant always started from the center of the screen before moving either to the left or right. The movement direction was selected randomly. E) After the movement, or after an equivalent idle period in the stationary condition, the screen went dark and a second bell sound was played. When the participants heard the second bell, they placed the red dot at the remembered target location using the keyboard (pressing the up, down, left, or right arrow keys moved the red dot on the screen as in the training). When the red dot was at the position where the participant thought the target was, they pressed the return key to indicate their choice. The red dot location was recorded as indicating the participant's remembered target location.

## Data analysis

In the data cleaning process, we treated any error value deviating more than 2.5 standard deviations from the mean (for each condition) as a mistake by the participant (e.g., missing the target during presentation or forgetting the position during movement) and removed those points before calculating the average error for each participant. If all the responses for any condition were removed, that condition for that participant was removed from the analysis. The same process was used in all three experiments.

The data from the two observer start positions (Left and Right) were brought to the same side where the positive (+) values represented errors in the direction towards the opposite side of the observer position for Stationary condition (e.g., errors to left were positive if the observer was on the Right side) and in the direction of the movement for the Visual Motion condition (e.g., errors to the left were positive if the observer was moved from right-to-left). Only horizontal errors were considered since the movements were in lateral directions only which were calculated by subtracting the actual target position from the remembered target position.

Exp 1: The data cleaning process removed 53 data points from the total data points (2.35%). No participant was removed from the analysis. A repeated-measures ANOVA with a Greenhouse-Geisser correction was conducted to compare errors, evaluating the effect of initial observer position (Left and Right), motion (Stationary and Visual Motion), and target position (brought to the same side).

Exp 2: The data cleaning process removed 131 data points from the total data points (3.40%). As a result, one participant had to be removed from the analysis due to having an incomplete data set leaving 23 participants for the analysis. A repeated-measures ANOVA with a Greenhouse-Geisser correction was conducted to compare errors, evaluating the effect of initial observer position (Left and Right), motion (Stationary, Visual Motion, Physical Motion, and Full Motion), and target position (brought to the same side).

Exp 3: The data cleaning process removed 166 data points from the total data points (3.22%). As a result, three participants had to be removed from the analysis due to having an incomplete data set leaving 23 participants for the analysis. We performed a two-factors repeated measure ANOVA with a Greenhouse-Geisser correction to compare the effect of motion conditions (Stationary, Short Travel, and Long Travel) and target position (collapsed between the observer positions) on the errors in participants' remembered target positions and used a family-wise alpha of .05.

**Updating ratio.** Klier et al. [11] considered perfect updating to occur when the observer's remembered target position after a translation (observer moving from an initial to a final position) equaled the remembered target position when stationary at the observer's final position (refer to their Fig 2 in [11]). Based on this assessment, they computed an "updating ratio"–the ratio between the estimated target position after the translation and the distance between the estimated positions when they were stationary at the initial and final observer positions–in order to evaluate people's updating. Klier et al. [11] measured gaze (i.e., version and vergence angles). In our experiments, instead of tracking the participants' eyes, we recorded the errors participants made when pointing at the remembered target positions. To use an updating index we converted our errors into pointing angles from the observer (see Fig 3), and then computed the "updating ratio" using the following formula:

$$\text{Updating Ratio} = \frac{PA_{MC} - PA_{I.SC}}{PA_{F.SC} - PA_{I.SC}} \tag{1}$$

where $PA_{MC}$ is the mean pointing angle of their remembered target position at the final position

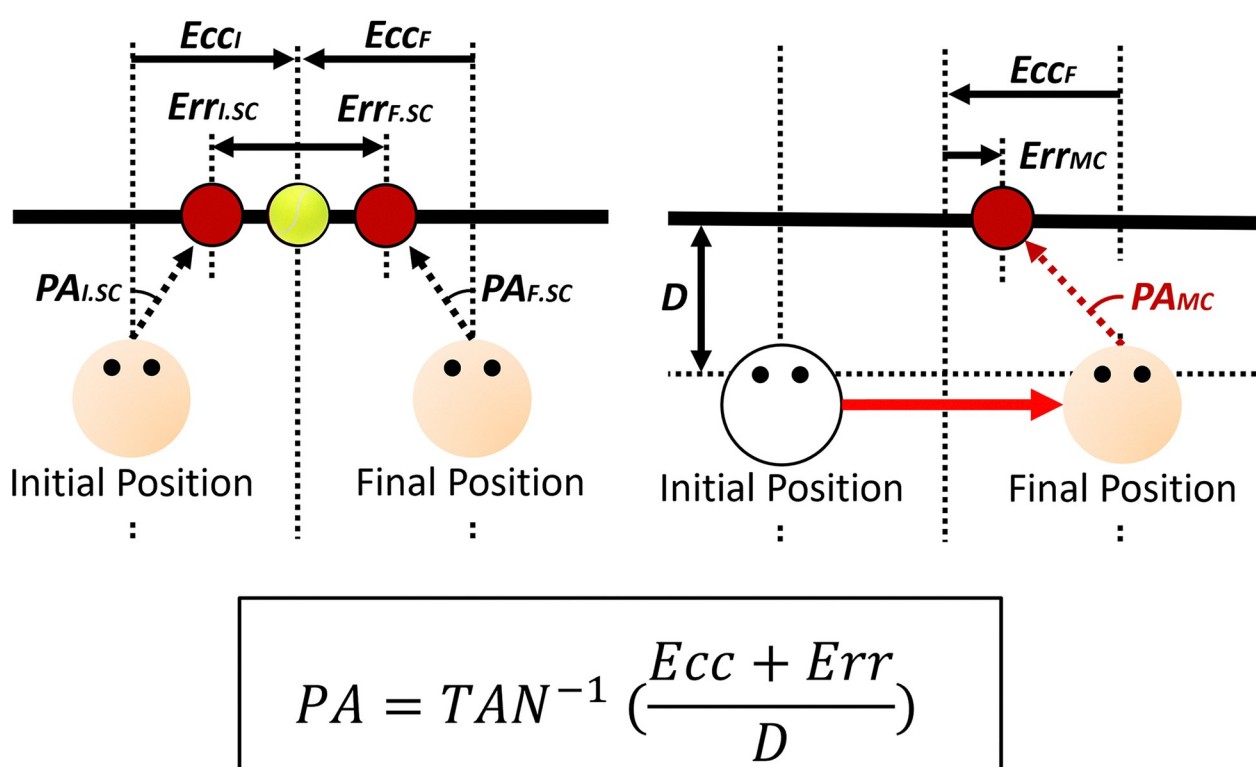

**Fig 3. Calculating pointing angles from the errors observer made when pointing at the remembered target positions.** $Ecc_I$ = Initial target eccentricity, $Ecc_F$ = Final target eccentricity, $Err_{I.SC}$ = Error when stationary at the initial observer position, $Err_{F.SC}$ = Error when stationary at the final observer position, $Err_{MC}$ = Error after being moved to the final position, $PA_{I.SC}$ = Mean pointing angle when stationary at the initial observer position, $PA_{F.SC}$ = Mean pointing angle when stationary at the final observer position, $PA_{MC}$ = Mean pointing angle after being moved to the final position, $D$ = Distance between the observer and the screen.

after being moved, $PA_{I.SC}$ is the mean pointing angle of their remembered target position when stationary at the initial observer position (right side if the translation was from right-to-left and vice-versa), and $PA_{F.SC}$ is the mean pointing angle of their remembered target position when stationary at the final position (left side if the translation was from right-to-left and vice-versa). Fig 4 shows the significance of this updating index. Perfect updating corresponds to an index of 1 where participants accurately remember the locations of targets in the world. An index of 0 indicates no updating at all in which the observer responds as if they had not moved at all.

## Results

Participants placed a red dot on a virtual screen simulated in VR to indicate a target's perceived position after various sideways movements. Errors were calculated by measuring the deviation between the remembered and actual target positions.

### Experiment 1

For experiment 1, the targets were between -0.75 m to +0.75 m (7 targets; distance between targets = .25 m) and movement was visually simulated at ±1 m. We conducted a repeated-

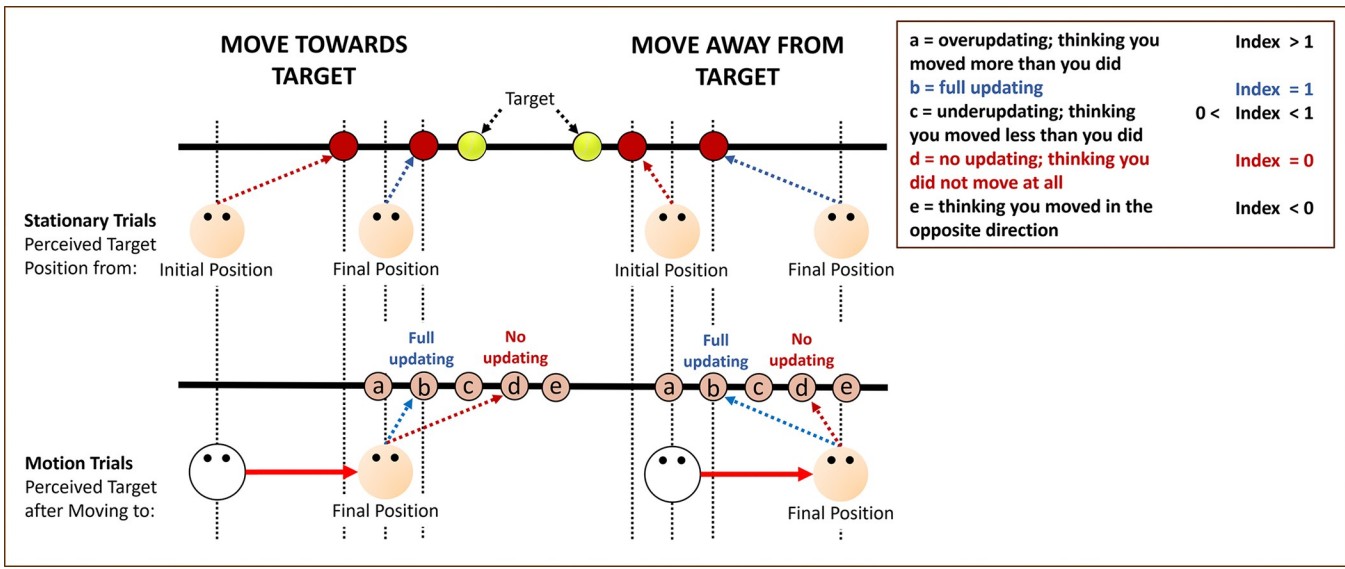

**Fig 4. Showing how different errors result in different "updating indices".** The position of the target presented is shown as the yellow tennis ball. The remembered target position after it disappeared when the observer was at the initial position (I) and at the final position (F) are shown as red balls. Then after a movement to the right (away from target–left panel, or towards target–right panel), possible errors are illustrated. The dashed blue line indicates full updating, i.e., remembered target position after moving is the same as position the observer would have remembered if they were stationary at the final position (b). If instead the observer positions the target at other locations they correspond to over-updating (a), too little updating (c), no updating at all (dashed red line) (d), or thinking the observer moved in the opposite direction (e). The index value for each of these outcomes is indicated in the legend.

measures ANOVA with Greenhouse-Geisser correction to compare errors, evaluating the effect of observer position (Left and Right), motion (Stationary and Visual Motion) and target position (collapsed between the observer positions). An, lysis revealed no significant differences in mean errors due to the initial observer position, $F(1, 19) = 0.053$, $p = .821$, $\eta2 < .001$. However, there was a significant main effect of motion, $F(1, 19) = 16.085$, $p < .001$, $\eta2 = .213$, where the errors in Visual Motion condition shifted after moving (0.2 m on average) in the direction of travel compared to Stationary condition. A significant main effect of target position was also found, $F(1.999, 37.973) = 37.771$, $p < .001$, $\eta2 = .234$. These main effects were validated with a significant interaction (motion × target position), $F(2.619, 49.769) = 3.013$, $p = .045$, $\eta2 = .012$.

Post-hoc evaluation of paired t-test, with Holm-Bonferroni correction, showed that the errors were larger for more eccentric targets in general. The errors significantly differed between the Stationary and the Visual Motion conditions for the targets at -0.25 m ($p = .008$), 0 m ($p = .004$), and at 1.25 m ($p = .011$) as represented in Fig 5C (these are equivalent to the targets at 0.75 m, 0.5 m and -0.75 m in Fig 3A, and -0.75 m, -0.5 m and 0.75 m in Fig 5B), the targets outside or at the end of the travel range (between 0 m and 1 m). In the Visual Motion condition (where sideways motion was simulated visually), the errors differed significantly ($p < .001$) between the targets at 0 m and 0.25 m, as represented in Fig 5C (these are equivalent to the targets at -0.5 m and -0.25 m in Fig 5A, and 0.25 m and 0.5 m in Fig 5B). This difference effectively separates the errors made for the targets within and beyond the travel range (between 0.25 m and 1.25 m, shown as shaded areas in Fig 5C) from those at the initial position (0 m) and away from the final position (-0.25 m). These results show that updating depended on target eccentricity as well as on the travel range and those targets within the range may have been updated more accurately (with smaller errors) compared to those outside the range, but not always (see Figs 6 and 7). However, motion was only visually simulated.

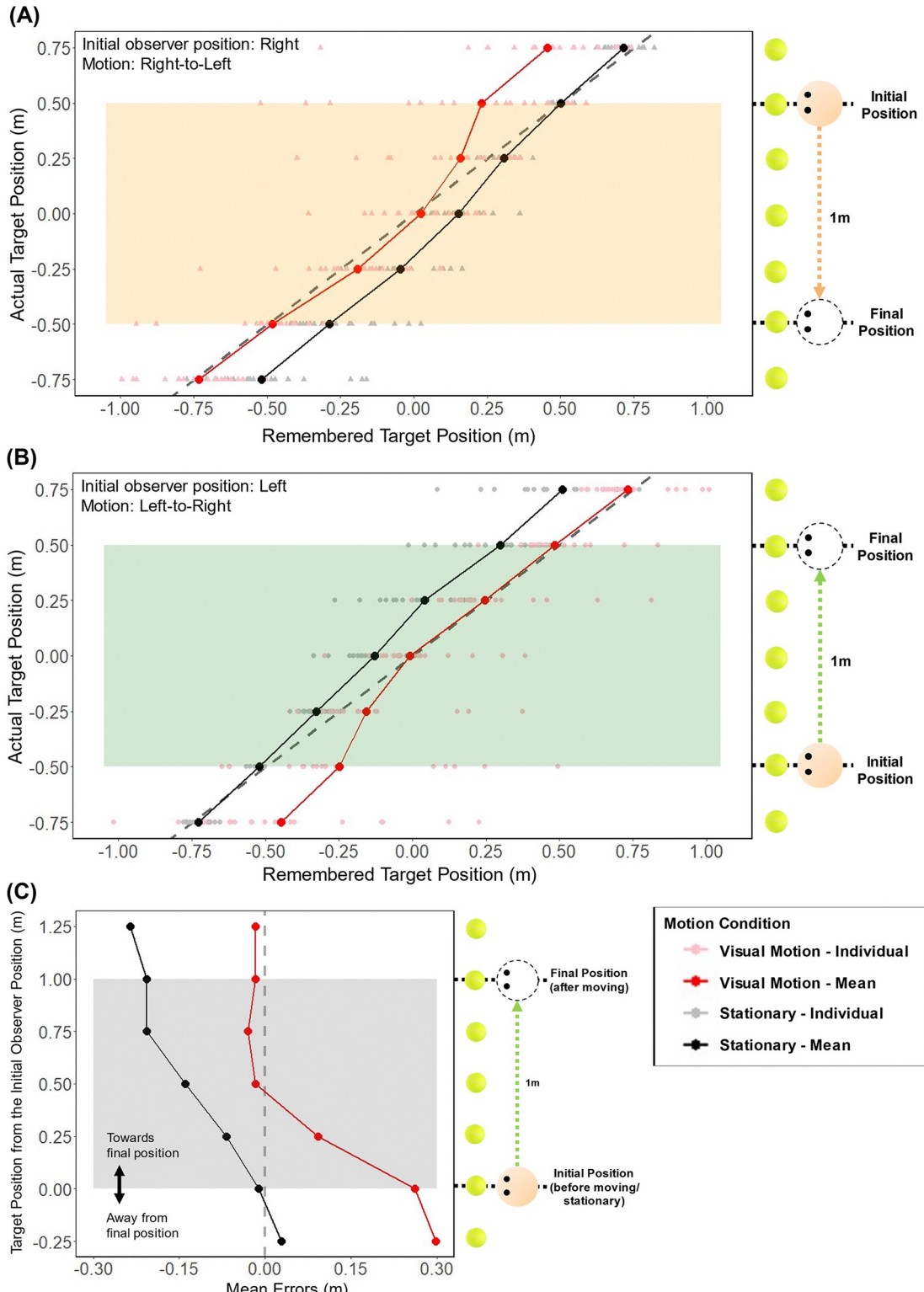

**Fig 5. Results of Exp 1.** (A) and (B) indicate the target positions participants indicated as they remembered them on the screen after Stationary (Black) and Visual Motion (Red), where (A) the observer was initially on the left side moving right, and (B) the observer was initially on the right side moving left. The lighter colored dots represent remembered positions from each participant (N = 20) and the darker colored dots and lines represent the means. The dashed line represents true target positions, therefore the deviation from the dashed line to the remembered target position is the error made. For both x- and y-axes, positive

(+) represents right, and negative (-) left direction from the center of the screen. (C) indicates the average errors for each target for each condition. For both x- and y-axes, positive (+) represents towards the moving direction, and negative (-) away from the moving direction in the moving condition (for stationary condition, the directions were kept the same for consistency). The shaded area (A–Orange, B–Green, C–Grey) contains the targets within the start and the end positions (from -0.5 m to +0.5 m) between which the observer traveled in the Visual Motion condition.

## Experiment 2

For experiment 2 we added physical motion created with a MOOG motion platform. The targets were between -0.46 m to +0.46 m from the center of the virtual display and motion was ±0.46 m. We conducted a repeated-measures ANOVA with Greenhouse-Geisser correction to compare errors, evaluating the effect of observer position (Left and Right), motion (Stationary, Visual Motion, Physical Motion, and Full Motion) and target position (collapsed between the observer start positions). As in Exp 1, there were no significant differences in mean errors due to the initial observer position, $F(1, 22) = 0.208$, $p = .653$, $\eta2 < .001$, and there was a significant main effect of target position, $F(1.307, 28.744) = 36.676$, $p < .001$, $\eta2 = .210$. However, there was no significant main effect of motion condition, $F(1.311, 28.842) = 1.644$, $p = .213$, $\eta2 = .024$, and no significant interaction between motion and target position, $F(3.002, 66.054) = 1.911$, $p = .136$, $\eta2 = .014$.

Further evaluation of the effect of target position revealed that the errors were significantly different ($p \leq .001$) between the targets from -0.23 m to 0.46 m (Between targets -0.46 m and -0.23 m, and between targets -0.23 m and 0 m in Fig 6A, between targets 0 m and 0.23 m, and between targets 0.23 m and 0.46 m in Fig 6B). These results confirm that the errors depend on the target eccentricity. However, unlike in Exp 1, the errors did not differ between the Stationary and the Visual Motion conditions ($p = .887$).

## Experiment 3

Both Exps 1 and 2 were constrained in terms of the eccentricities of the targets and the self-motion used. In Exp 3, we increased the range of target eccentricities to ±1.15 m and compared the effect of short (±0.46 m) and long (±1 m) translations. Fig 7 shows the participants' responses under each condition. An ANOVA revealed there was a significant main effect of motion condition, $F(1.451, 31.917) = 15.896$, $p < .001$, $\eta2 = .163$, where errors in both Short (M = 0.102 m, SD = 0.291 m) and Long travel conditions (M = 0.266 m, SD = 0.432 m) significantly differed from Stationary (M = -0.015 m, SD = .210 m), $p < .001$. A significant main effect of target position was also found, $F(2.085, 45.875) = 26.165$, $p < .001$, $\eta2 = .184$. There was a significant interaction between travel distance and target position, $F(7.186, 158.097) = 3.950$, $p < .001$, $\eta2 = .018$.

Post hoc evaluation of paired t-test, with Holm-Bonferroni correction showed that the mean errors made for each target position significantly differed between motion conditions, where the errors shifted after moving compared to Stationary in the direction of travel for both Short ($p = .039$) and Long ($p < .001$) Travel conditions (0.1 m for short and 0.27 m for long travel on average). The errors also differed between the travel distances ($p = .002$), in which longer travel distances resulted in larger errors.

## Comparing error differences between Experiments 2 and 3

Exps 2 and 3 showed conflicting results where errors in remembered target positions after a passive visual motion were different from those after being stationary in Exp 3, but not in Exp 2. Fig 8 shows participant errors for the targets that were equivalent between the two experiments, i.e., targets between -0.23 m and +0.69 m in Stationary and Visual Motion (travel

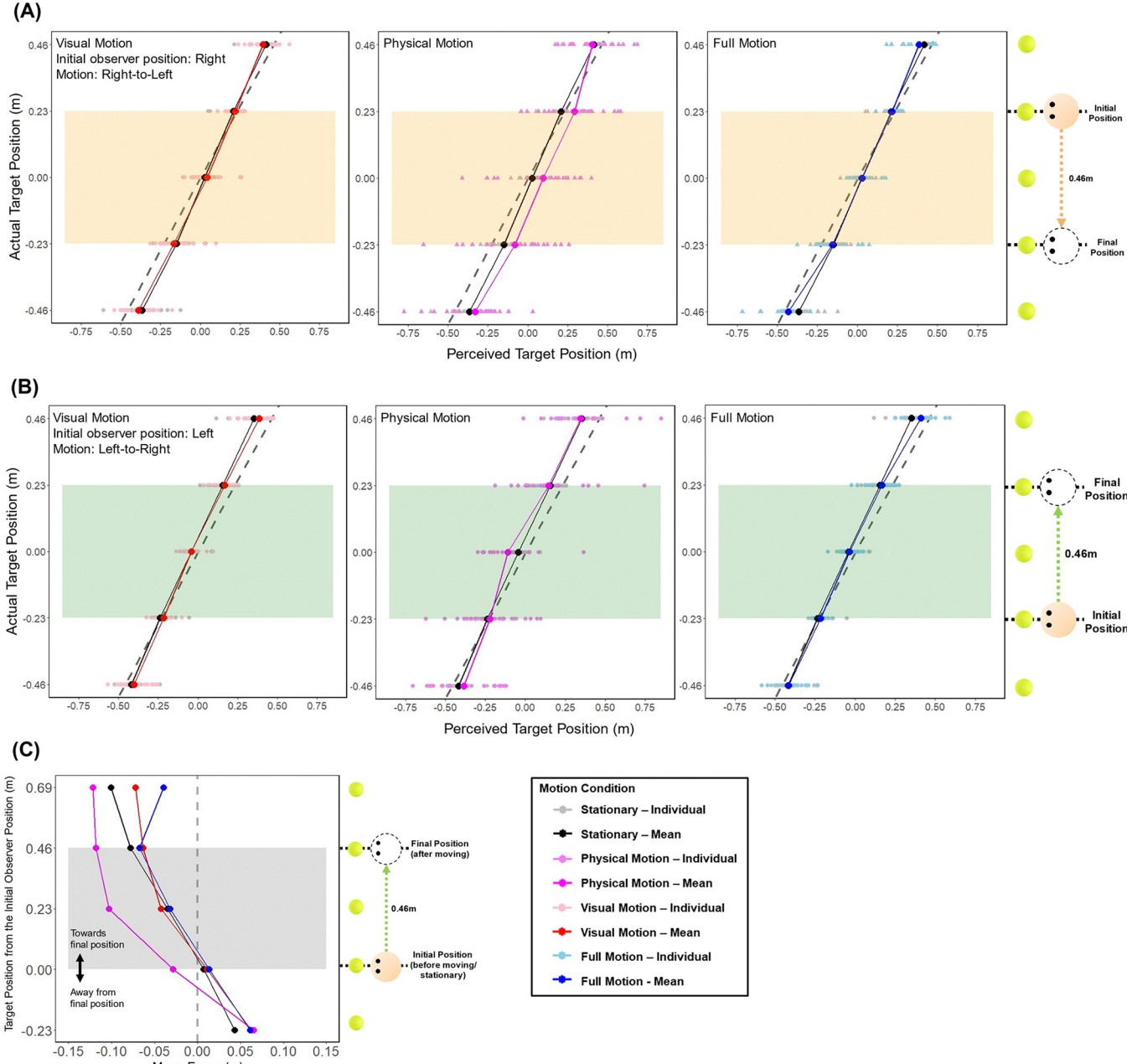

**Fig 6. Results of Exp 2.** (A) and (B) indicate the target positions participants indicated as they remembered them on the screen after Stationary (Black), Visual Motion (Red), Physical Motion (Purple) and Full Motion (vision and physical) (Blue) conditions, where (A) the observer was initially on the left side, and (B) the observer was initially on the right side. The lighter colored dots represent remembered positions from each participant (N = 23) and the darker colored dots and lines represent the means. The dashed line represents true target positions, therefore the deviation from the dashed line to the remembered target position is the error made. For both x- and y-axes, positive (+) represents right, and negative (-) left direction from the center of the screen. The orange and green shaded areas contain the targets within the start and the end positions (from -0.23 m to 0.23 m) that the observer traveled between in the motion conditions. (C) indicates the average errors for each target for each condition. For both x- and y-axes, positive (+) represents towards the moving direction, and negative (-) away from the moving direction in the moving conditions (for stationary condition, the directions were kept the same for consistency). The faded area (A– Orange, B–Green, C–Grey) contains the targets within the start and the end positions (from -0.5 m to +0.5 m) between which the observer traveled in the moving conditions.

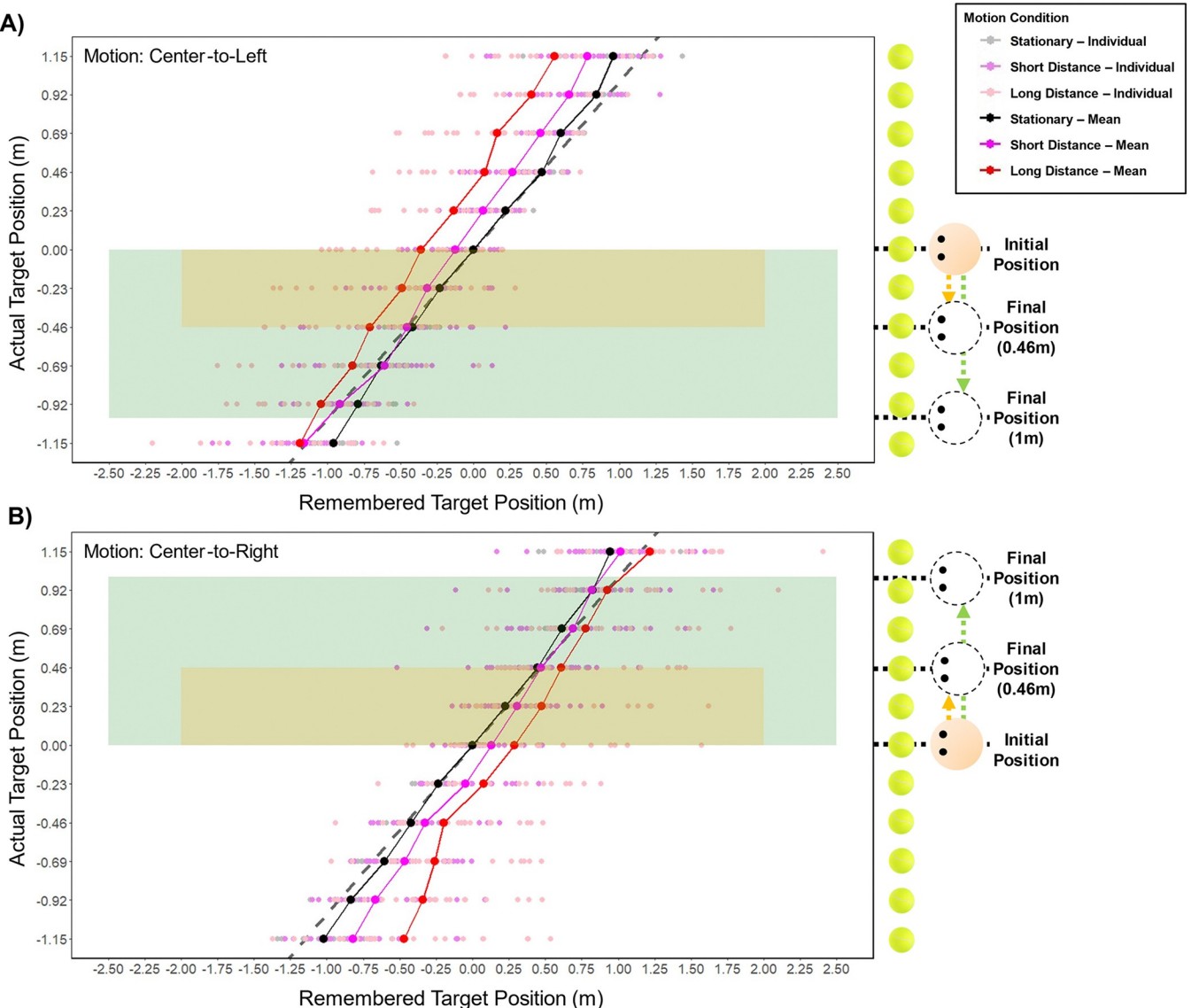

**Fig 7. Results of Exp 3.** The positions participants remembered the targets in for Stationary, Short (0.46 m) and Long (1 m) travel conditions. A) after moving from center to the left, B) after moving from center to the right. The lighter colored dots represent remembered positions from each participant (N = 23) and the darker colored dots and lines represent the means. The dashed line represents true target positions, therefore the deviation from the dashed line to the remembered target position is the error made. For both x- and y-axis, positive (+) represents right, and negative (-) left direction from the center of the screen. The orange and green shaded area contain the targets within the start and the end positions (orange from 0 m to 0.46 m, and green from 0 m to 1 m) between which the observer traveled in the moving conditions.

distance = 0.46 m, and travel time = 5 seconds) conditions. An ANOVA comparing the error differences (Error$_{stationary}$–Error$_{visual\ motion}$) revealed significant interaction between experiment and target position, $F(1.698, 74.698) = 3.639$, $p = .032$, $\eta 2 = .024$. Further evaluation of simple main effect showed that the error differences were significantly larger in Exp 3 for targets at -0.23 m ($p < .001$) and 0 m ($p = .006$).

The updating ratio for each target positions was computed for each target, using Eq 1 (see methods), in Exps 1 and 2. Fig 9 shows the updating ratio calculated for each target position from Exp 1 (A) and Exp 2 (B). These updating ratios between targets reveal a difference in participants' updating efficacy between the targets within the travel range (between targets at 0 m

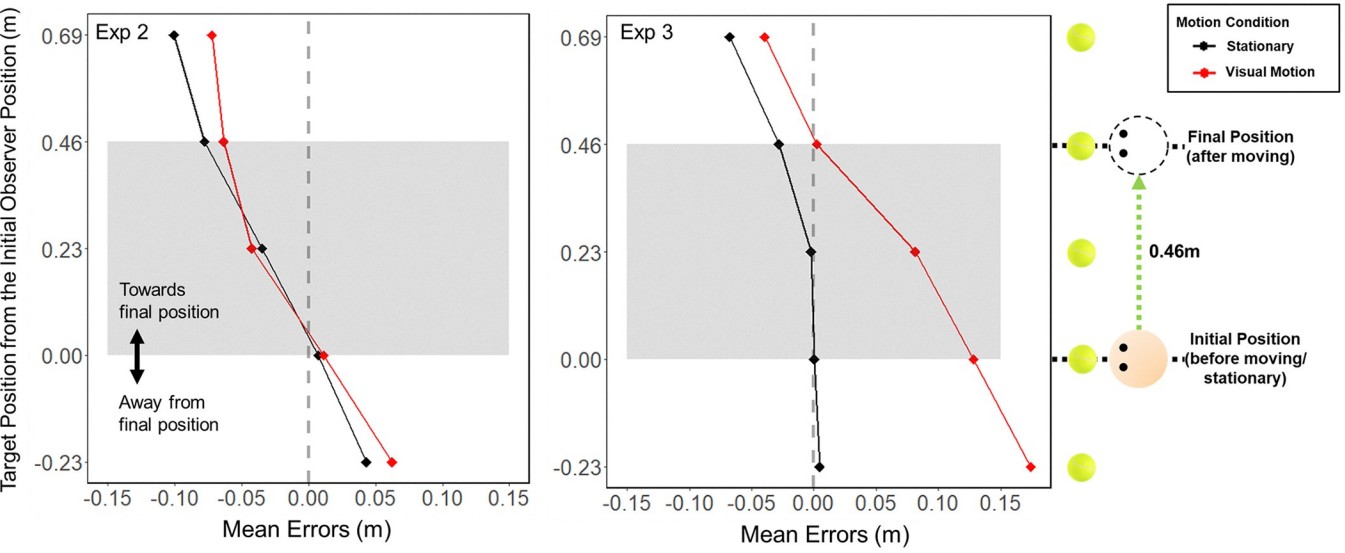

**Fig 8. Comparison between Exp 2 and 3.** Mean errors participants made for each target for Stationary (Black) and Visual Motion (Red) conditions for Exp 2 and the equivalent targets from Exp 3, where 0 on the vertical axis is the start position and collapsed across observer positions: Left and Right. For both x- and y-axis, positive (+) represents in the direction of the movement, and negative (-) the opposite direction. The faded area contains the errors for the targets from the start and the end position (from 0 m to 0.46 m) where the observer traveled in the Visual Motion conditions.

and 1 m for Exp 1, and 0 m and 0.46 m for Exp 2) and those outside this range. Updating ratios for Exp 3 could not be computed due to the difference in its method (observer's initial position always being in the center of the screen) from Exps 1 and 2.

## Discussion

By asking participants to remember the positions of targets following passive lateral translation or after a comparable period with no movement, we were able to assess the contributions of target eccentricity (Exps 1, 2, and 3), self-motion cues (Exp 2), and translation distance (Exp 3) on spatial updating performance. In all three of our experiments, participants underestimated target eccentricity even when they did not move at all, with the magnitude of their errors depending on target eccentricity. In Exp 1, these errors shifted after lateral translation in the direction of their translation. The errors participants made after moving differed systematically depending on target eccentricity where the errors were smaller for targets that became less eccentric as a result of the movement, and larger for targets that became more eccentric. These results support our hypothesis that updating depends on target eccentricity.

The errors represent a change in the observer's mental representation of target position after moving, i.e., updating errors. For Exp 2, we hypothesized that having more motion cues available during a lateral translation would make people more accurate at updating target positions due to a better perception of their self-motion. In the Full Motion condition, translation was performed on a MOOG moving platform which provides the full experience of self-motion (with both visual and physical motion cues) which we expected to help people to judge their self-motion more accurately [6, 7, 18]. A more accurate perception of self-motion should then lead to better judgement of travel distance and hence more accurate updating. Self-motion is typically overestimated with only visual or physical motion cues with larger overestimation with only physical motion cues [7]. We expected updating errors to be smallest in the Full Motion condition, followed by the Visual Motion condition, then largest in the Physical Motion condition. However, the errors after being translated laterally with more motion cues

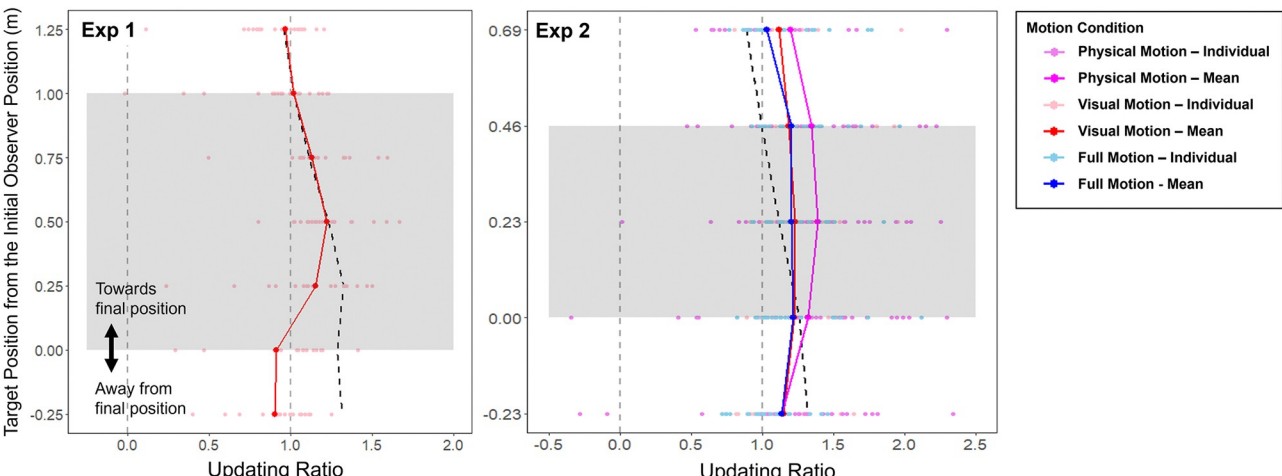

**Fig 9. The updating ratio for each target position.** Exp 1: with 1m translation–visual motion (Red) without any physical motion cues. Exp 2: with 0.46 m translation–three motion conditions: visual motion (Red), physical motion (Purple), and full motion (Blue). The lighter colored dots are updating ratios from each participant and the darker colored dots and lines are the means. The vertical grey dashed lines represent perfect updating at 1 and no updating at 0. The black dashed line represents mean updating ratios if participants had correctly remembered target positions after moving. The grey shaded area represents the start and end position through which the observer traveled (Exp 1 –from 0 m to 1 m, Exp 2 –from 0 m to .46 m) in the moving condition.

(Full Motion condition–both visual and physical motion cues) did not significantly differ from those conditions with fewer motion cues (Visual Motion and Physical Motion conditions), hence our hypothesis was not supported by our results. Moreover, the errors without translation did not significantly differ from the errors with translation for any of the motion conditions, i.e., participants' remembered target positions were the same with or without lateral translation, which suggest there was no error associated with moving (updating error).

There were a few differences in Exp 2 compared with Exp 1 which might have contributed to this change in updating errors: a) Shorter travel distance and travel time, and b) Exposure to the test room and MOOG. Due to the physical limitation of the MOOG, the travel distance was reduced to 0.46 m in Exp 2 compared with the 1m used in Exp 1. The travel time was also shortened to keep the experiment under 2 hours (7 seconds in Exp 1 and 5 seconds in Exp 2). The results in Exp 3, testing the effect of travel distance, the errors shifted even with the reduced 0.46 m translation, although the shift was smaller than for the 1.0 m translation. Therefore, it is unlikely that shorter travel distance is the reason for the error shifts after translation going away in Exp 2. Based on these results, it appears that exposure to the MOOG reduced participants' updating errors in Exp 2. We will now discuss how each factor may impact people's updating in more detail.

### Errors in localizing targets while stationary

Our results (see Figs 3–5 - Stationary) where people simply tried to remember objects' locations, were compatible with the past studies [16, 17] as we hypothesized. In the past studies, the underestimation of target eccentricity was assessed by measuring observers' saccades or reaching to targets, making it unclear whether the undershoot was due to the misperception of target eccentricity or errors in eye/arm movements. In Exp 3, where the participants used buttons instead of pointing at target locations, they still underestimated target eccentricities indicating that the errors most likely arose from misperceived target position. In addition, unlike

past studies, the present study used targets beyond the observers' arm's reach demonstrating for the first time that the undershooting effect extends to the targets that are not reachable.

## Updating errors following visual lateral translation

In Exp 2 the updating errors after visual lateral translation were much smaller (Exp 2: 0.012 m on average) than they were in Exp 1 (Exp 1: 0.2 m on average) which resulted in significant difference between the errors with and without visual translation in Exp 1, but not in Exp 2. Fig 6 compares the error results from Exp 3 (where the participant was sitting on a MOOG motion platform, i.e., knowing that the chair could move) to those from Exp 2 (where they were sitting in a regular lab, i.e., knowing that the chair could not move) for the targets that were equivalent. This comparison reveals that the lack of updating errors in Exp 2 may be due to differences in the rooms in which they were tested. In Exp 2, participants were tested in a physical room which the virtual environment was designed to mimic (although the experiment was done wearing an HMD participants had a clear view of the room and the motion platform on which they were going to sit while they were preparing for the experiment). Distance compression in VR can be improved when participants are previously exposed to a real-world environment prior to testing in the identical VR environment [19, 20]. Our participants' exposure to the MOOG may have helped them perceive the VR environment accurately thus reducing the underestimation of target distance and resulting in a more accurate representation and scaling of the virtual environment. Additionally, sitting on a moving platform in Exp 2, even when it did not actually move (visual motion condition), has been shown to enhance vection [3, 21]. Such an enhancement would then lead to more accurate perception of travel distance compared with Exp 1 and 3 where the participants sat on a chair that they knew was not able to move. Based on these considerations, it seems that even potential physical motion cues play an important role in spatial updating. Even when there is no actual physical motion involved, knowing that one can move can still improve one's perception of visually induced self-motion and make spatial updating more accurate. If exposure to the motion platform and physical motion cues led to more accurate perception of self-motion and travel distance, then the updating errors due to underestimated travel distance would have been effectively removed or reduced in Exp 2.

## Updating the position of objects that cross the midline

When moving laterally, the eccentricity of a laterally displaced target changes. For example, in Exp 1 participants were translated laterally by 1 m which resulted in the egocentric target positions being shifted by 1 m (e.g., target at -1 m eccentricity ended up being at 0 m after a +1 m translation). People already underestimate the eccentricities of laterally displaced target objects even if they do not move [16, 17]. If people were to make the same underestimation after translation, with the new updated target eccentricities, the errors people make would shift corresponding to each target's new position in the environment.

The updating ratios in Fig 9 show that the targets within the travel range had larger updating ratios compared to those outside the range. That is, updating efficiency seems to be moving-range dependent—its effectiveness depending on whether the target is within or outside of the movement range. The targets within the moving range cross from one hemifield to another after the participant's movement (Fig 10). Theories of how the brain codes space, specifically information from the two sides of space, have been informed by studies of spatial neglect which suggest that information from each hemifield of egocentric space (relative to the body) are coded independently [22, 23]. So, when remembering the position of an object on the left before a movement to the left that goes past the object and causes its representation in memory

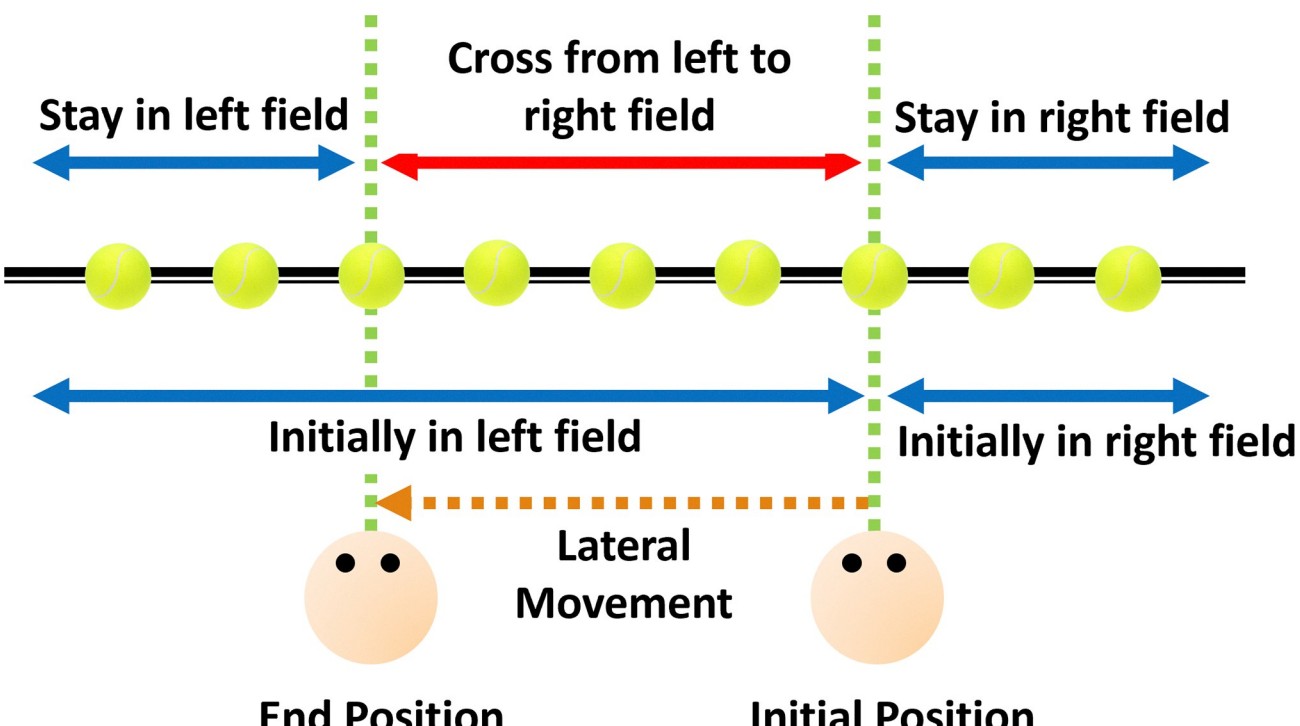

**Fig 10. Change of the target visual field after observer lateral movement based on the target eccentricity.** Targets within the range of motion (indicated by the red arrow) are initially in the observer's left field but switch to the observer's right field as a result of the movement. Targets outside this range merely move within their original fields.

to shift from left to right relative to the midline, would require it to shift across into the representation of the other hemifield. The cross-hemispheric switch of a target initially in one hemisphere transferred to the other hemisphere after eye movement [24, 25], as well as after full body movement [9, 10] has been observed in human parietal cortex. This shifting process was shown as a continuous updating in the brain with a slow eye movement [26]. We suggest that it is this transfer which causes the larger updating ratios.

Why might targets that cross the midline be updated differently compared to those that do not cross into the other hemifield? Humans generally do not move laterally. When they do, however, it is typically when trying to stop something from going past them (e.g., a goalie blocking a ball) or actively pursuing a target (e.g., a hunter keeping track of a prey). In both cases, the target should be kept within the range of lateral movement for the person to be able to hone in on it. The classical pursuit strategy, where the pursuer moves directly toward the target, is inefficient compare to the interception strategy, where the pursuer moves to the estimated interception path with the target [27]. Over updating as a consequence of the open loop nature of predicting the position of an invisible target after a passive lateral movement [28] would mean that overcompensating for lateral movement could represent an efficient target catching strategy.

## Conclusions

When a person observes an object, their perception of its position is biased towards their straight ahead, leading to underestimating its eccentricity. Correctly gauging self-motion is important when updating an object's perceived position during and after movement. Familiarity with an environment and knowing whether a person is free to move (e.g., if sitting on a

moveable platform) both improve updating. The efficacy of spatial updating also depends on whether an object's position crosses the midline from one hemifield to the other after moving, which may provide an ecological advantage for more efficient pursuit of a target.

## Author Contributions

**Conceptualization:** John J. J. Kim, Laurence R. Harris.

**Data curation:** John J. J. Kim.

**Formal analysis:** John J. J. Kim.

**Funding acquisition:** Laurence R. Harris.

**Investigation:** John J. J. Kim.

**Methodology:** John J. J. Kim.

**Project administration:** John J. J. Kim.

**Resources:** Laurence R. Harris.

**Software:** John J. J. Kim.

**Supervision:** Laurence R. Harris.

**Validation:** John J. J. Kim.

**Visualization:** John J. J. Kim.

**Writing – original draft:** John J. J. Kim.

**Writing – review & editing:** John J. J. Kim, Laurence R. Harris.

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
