## [Decision Letter · Decision Letter 0]

9 Sep 2024

PONE-D-24-28073Updating the remembered position of targets following passive lateral translationPLOS ONE

Dear Dr. Kim,

Thank you for submitting your manuscript to PLOS ONE. After careful consideration, we feel that it has merit but does not fully meet PLOS ONE’s publication criteria as it currently stands. Therefore, we invite you to submit a revised version of the manuscript that addresses the points raised during the review process. **Three expert reviewers provide their detailed comments below. Please carefully address each of those comments, and emphasize more explicitly the importance of conducting the three experiments while clearly stating the respective hypotheses. In addition, please ensure that methodological decisions are clearly justified, and consider the suggested revisions about the figures. You will find further comments and suggestions in the section below.**

We look forward to receiving your revised manuscript.

Kind regards,

Dimitris Voudouris

Academic Editor

PLOS ONE

**Journal Requirements:**

These experiments were supported by a grant from Natural Sciences and Engineering Research Council of Canada RGPIN-2020-06093 to LRH. 

These experiments were supported by a grant from Natural Sciences and Engineering Research Council of Canada RGPIN-2020-06093 to LRH. We would also like to thank the Canada First Research Excellence Funded project Vision: Science to Applications (VISTA) for their support.

These experiments were supported by a grant from Natural Sciences and Engineering Research Council of Canada RGPIN-2020-06093 to LRH. 

Reviewers' comments:

Reviewer's Responses to Questions

**Comments to the Author**

1. Is the manuscript technically sound, and do the data support the conclusions?

Reviewer #1: Partly

Reviewer #2: Yes

Reviewer #3: Yes

2. Has the statistical analysis been performed appropriately and rigorously? 

Reviewer #1: Yes

Reviewer #2: Yes

Reviewer #3: Yes

3. Have the authors made all data underlying the findings in their manuscript fully available?

Reviewer #1: Yes

Reviewer #2: Yes

Reviewer #3: Yes

4. Is the manuscript presented in an intelligible fashion and written in standard English?

Reviewer #1: Yes

Reviewer #2: Yes

Reviewer #3: No

5. Review Comments to the Author

**Reviewer #1:** The present work investigates the ability to successfully update virtual target positions after visual/physical passive movements in healthy participants. The methodology appears sound. In three different experiments, participants are asked to judge the location of previously seen targets, while being stationary or after being moved (virtually with visual cues or physically through a motion platform).

Generally, this study has the potential to extend previous literature about spatial updating and the relationship between self-motion and the spatial processing of the surrounding environment, but some editing is required. Below are some major and minor comments and suggestions to foster publication.

Abstract:

- Authors claimed to measure errors before and after passive translations. This makes one think of a sort of pre- and post- movements comparison, but it does not seem to be the case. Is that correct?

Introduction:

- Page 3, lines 16-18: what is the reason for introducing the usual methodologies to prevent people from using visual cues in this context? It is true that in real life, visual motion signals are fundamental (lines 19-21), but in experimental procedures, there are reasons to select specific perceptual cues (e.g., to segregate acoustic or other senses effects on self-motion or spatial updating).

- Page 4, lines 2-4: can’t object distance change during rotations as well? I imagine, for example, rotating on a swivel chair in front of my desk: if I rotate by 180°, after rotation, I will not be able to grasp desk objects anymore, being them exactly behind me. Relative to the chair, the object probably maintained the position, but relative to my arm/hand, it actually changed, perhaps preventing me from grasping it. Am I missing what the authors are saying? Maybe it could be helpful to clearly define the “distance”.

- A stronger and more thorough introduction to the reasons for conducting three different experiments and the use of both visual and physical motion information could help the reader to immediately grasp the take-home message of this work. In addition, it is not very clear which are the specific hypotheses and which is the novelty relative to the cited previous literature (references 16,17). Some new aspects are mentioned in the discussion (page 22, lines 4-6), but it would make more sense to insert them in the introduction session.

Method/Procedure:

- Page 9, line 4: can the authors clarify the reason for adding randomly generated dots on the screen? Was the screen the virtually rendered one?

- What is the reason for changing the range of target displacements, increments, duration, and velocity of motion information across the experiments? Why did the authors decide to make the dots blink in Experiment 3?

- I do not think I got the virtual visual motion information: was it consistent with a real movement, or did it consist of moving the surroundings?

- How many trials for each condition in each experiment?

Data analysis:

- Page 13, line 10: Why did the authors consider errors deviating more than 2.5 SD only as a participant mistake and not as general variability?

- If the data from the two observer positions were brought to the same side, why one of the factors of the ANOVAs is the observer position? Cannot be better to consider observer position levels like “same side vs different side” of the target instead of left and right?

Results:

- Experiment 1: In Figure 3C, it seems that Visual Motion condition favor participants to reduce errors if compared with Stationary condition. If this is correct, the authors could mention this effect more explicitly, to clarify the direction of the reported difference between conditions.

- Figure 5A and B seem inverted: in Figure 5A, the participant seems to go from the center to the left; in Figure 5B from center to the right. If that’s correct, caption should be corrected.

Discussion:

- I believe that a wider summary of the results is needed before the paragraph (page 21, lines 2-11) about Experiment 2 results.

- Page 22, lines 15-17, “In Exp 1 these updating errors resulted in smaller errors for targets that became

less eccentric as a result of the movement, and larger errors for targets that became more

eccentric”: in Figure 3C, it seems the opposite. The red line for Visual Motion condition is closer to 0 as the position of the target is further from the initial position of the participant.

- Page 24, lines 3-6: The idea that potential physical motion could have played some role in spatial updating is not totally convincing to me. The actual physical movement did not improve participants’ performance (Exp 2), so why should the potential movement have a positive effect?

- The discussion section about the “updating ratio” (from page 25, line 7) seems to me more appropriate for a results section than the discussion.

- Page 26, line 3: “1. 0”: typo

- Page 29, line 10: the example should be about humans again, for better coherence.

- Generally, I would suggest to restructure the discussion to favor a better comprehension of the results as follows: Summary of general results � Results from Exp 1 and implications � Results from Exp 2 and implications � Results from Exp 3 and implications � general implications and interpretations. In the current discussion, the updating ratio computation and the first paragraph about Exp 2 come across as confusing.

**Reviewer #2: **This manuscript is an important advancement in the understanding of the mechanisms of spatial updating, especially as they relate to lateral translation movements. The authors conducted three experiments in a virtual reality setting, where they manipulated in various ways, the amount of (un)certainty about self-motion.

Major comments:

1. The introduction needs a bit more expansion, especially in terms what the previous studies that have investigated spatial updating with rotational movements and how the authors might envision spatial updating will change for lateral movements. This will help to motivate the study and the specific hypotheses more.

2. The authors indicate that they conducted three experiments and have one main hypothesis outlined in the introduction. The authors should tie each experiment to the hypothesis or expand and indicate why each of the three experiments was needed.

3. There are some details in the Methods section that need to be added (e.g., the number of trials in total; whether the conditions blocked or mixed; the number of repetitions per trial type; the rooms that the experiments took place in; etc.), as well as motivation behind their approaches (and changes across the experiments). For example, why were the distances between the targets chosen, and changed, in the three experiments? How do each of the changes contribute to answering how spatial updating for lateral movements? These specific methodological motivations are important for understanding how each experiment is motivated and in what way it contributes to answering the overarching question of the manuscript.

4. For experiment 2, could the authors indicate the specific progression of information for the different conditions tested and the effect on spatial updating. For example, was the physical and visual motion condition expected to reduce uncertainty around self-motion the most and therefore, produce the smallest errors; if so, was the next best condition considered to be the physical only condition, etc.? How do the results compare to these expectations? These are very briefly covered in the discussion.

Minor point:

1. Could the authors provide an explanation as to why the experiments were conducted in virtual reality settings and not in the actual environments?

2. Why were controllers used in the first two experiments, but not in the third? How does this help to disentangle the spatial updating process for lateral movements?

**Reviewer #3: **Kim and Harris report that during lateral translation, subjects systematically underestimate target eccentricity, particularly for more eccentric targets, with errors influenced by whether the target crosses the observer's midline. These errors likely reflect changes in spatial updating tied to perceived self-motion and reconstructing target positions across visual fields. While the results are interesting, the manuscript would benefit from clearer data representation, both in the text and figures. Additionally, the writing (several errors) could be substantially improved (could be more compact) for better clarity and impact.

Major issues:

EXPERIMENT 1:

• The text is not clear. What is the spacing between targets? I agree that it is mentioned in the “methods’ but it is good to remind the reader of the relationships between target positions, travel range, and motion simulation. My major concern here is the overinterpretation of the main effect (page 15, lines 15-16) as reported by a significant interaction (motion and target) with p = 0.045. I suggest emphasizing the small effect size in the interaction, and a much clearer explanation for spatial updating. What are the data points to the extremes of these lines (- 0.75 and 0.75), though the translation is between – 0.5 and 0.5? Since all the subjects are right-handed, was there any hemi-field effect (it is reported that errors increased with eccentricity, it seems there is no hemifield effect but could be mentioned / investigated further)? Can the authors indicate the significance on the graph? I think it is more informative and intuitive to switch the x and y axis. The data should be plotted: Remembered target position as a function of actual target position. That way the cartoon on the right will be aligned on the x axis, thus better representing the horizontal translation (and motion). Along the same lines, instead of a columnar organization for the figure, the data could be represented in a row.

• This sentence is not clear (Page 15, lines 18-21): “In the Visual Motion condition (where sideways motion was simulated visually), the errors significantly differed (p < .001) between the targets at 0m and .25m (Targets at -0.5m and -0.25m in Fig 3A, Targets at 0.25m and 0.5m in Fig 3B).” Please clarify.

• The statement (lines 2-4, page 16) is vague: ‘suggesting that updating may be dependent on target eccentricity as well as the travel range since those targets within the range may be updated differently compared to those outside the range.’’ What does ‘differently’ mean?

• Accordingly, the spatial updating conclusion should be clearly derived. I agree that the platforms are being passively moved, given that the neck was restrained, the head did not contribute to the updating mechanism(s). Can the authors speculate in discussion how the head may contribute to it (thus reaching) in the real world?

EXPERIMENT 2:

• What led to Experiment 2 from Experiment 1? The text should begin with a reason for Experiment 2. What did the authors choose a different range (-.46 - .46) in this exp compared with Exp 1? Figure 5 seems to explain this but the decision for this range should be mentioned here too. It is also discussed in ‘discussion’. Despite that, it should be mentioned here.

• motion was 13 ±0.46m. Not clear. Is it motion (should be in m/s ) or initial position (as I see in table 1?)

• Why does Fig 4C precede 4A and 4B? The text should follow systematically with figures. I can see the effect of the target eccentricity on errors in Fig c, but the authors do not do justice to the figures by explaining them well. Furthermore, how do the authors explain the larger spread of errors during physical motion than the other two conditions? In fact, it also seems evident in 4C. I also suggest plotting these errors with all the data points.

• The authors mention (Page 18, line 1): “However, unlike in Exp 1, the errors did not differ between the motion conditions”. How do they exactly mean it? Any stats?

• Please look at my comments on EXP 1 for figures (you can go along those lines). The authors should describe visual (left), physical (middle) and full motion(left) better in the text. Again, the data can be plotted as suggested in 1. Why does the label change here to ‘Perceived target location’? Keep it consistent.

EXPERIMENT 3: See my earlier comments, however, to plot the difference between the short and long distances, the graph doesn’t help much. Can the authors think of any other way of plotting the data? Moreover, the tests they performed are rather opaque to the reader.

• Here in the text: Post hoc evaluation of paired t-test, with Holm-Bonferroni correction showed that the mean errors made for each target position significantly differed between motion conditions, where the errors shifted after moving compared to Stationary in the direction of travel for both short (p 20 = .039) and long (p < .001) travels. The errors also differed between the short and long travel 1 distances (p = .002), in which longer travel distances resulted in larger errors.

• How do confirm the above effect in both directions? Are the data pooled (normalized) for both directions? Please clarify.

• Can the authors plot the data (errors) in the spatiotemporal domain: errors as a function of the reaction time?

• I suggest moving the figures from discussion into the results section. This will help to further streamline the discussion section (to make it shorter).

• The authors could discuss a neural mechanism involving the integration of spatial information at the neural across brain areas. While Dash et al. (2015) [Continuous Updating of Visuospatial Memory in Superior Colliculus during Slow Eye Movements] focus on the gaze system, the authors might suggest something along these lines for the reaching system. In other words, how the gaze system may be involved in updating reaching plans, while also discussing perception of self-motion.

Other issues:

• Introduction: Needs to be streamlined (and could be more compact) with clear hypotheses. Should briefly mention results and conclusion.

• Discussion: Needs to be substantially shortened.

• Line 13: omit ‘of course’ and several other instances where the language is not scientific (e.g., obviously).

• Page 21, lines 14-15. These results could be mentioned in the results section.

• Remove contractions such as: ‘didn’t’ should be replaced with ‘did not’.

• What is the size of the virtual tennis ball?

• What does a passive movement mean? A little detail would be beneficial for the reader.

• Page 6, line 14: what is FOV?

• Make sure the units are represented consistently and with a ‘space’ between the value.

• Figure 2 (E): In the figure I see the background white rectangle. If this is the case, the participants can use it as an allocentric landmark. Did the experiments simulate darkness? I understand that there was some virtual background to the scene. What us the color of the fixation dot and other features during presentation?

• At several instances the text should read ‘compared with’ instead of ‘compared to’. These two have specific meanings.

6. PLOS authors have the option to publish the peer review history of their article (what does this mean?). If published, this will include your full peer review and any attached files.

Reviewer #1: No

Reviewer #2: No

Reviewer #3: No

---

## [Author Response · Author response to Decision Letter 0]

9 Oct 2024

Response to Reviewers:

Reviewer #1: The present work investigates the ability to successfully update virtual target positions after visual/physical passive movements in healthy participants. The methodology appears sound. In three different experiments, participants are asked to judge the location of previously seen targets, while being stationary or after being moved (virtually with visual cues or physically through a motion platform).

Generally, this study has the potential to extend previous literature about spatial updating and the relationship between self-motion and the spatial processing of the surrounding environment, but some editing is required. Below are some major and minor comments and suggestions to foster publication.

Abstract:

- Authors claimed to measure errors before and after passive translations. This makes one think of a sort of pre- and post- movements comparison, but it does not seem to be the case. Is that correct?

-We measured errors with or without movement (i.e., passive translations). We have now corrected this statement in the abstract. 

Introduction:

- Page 3, lines 16-18: what is the reason for introducing the usual methodologies to prevent people from using visual cues in this context? It is true that in real life, visual motion signals are fundamental (lines 19-21), but in experimental procedures, there are reasons to select specific perceptual cues (e.g., to segregate acoustic or other senses effects on self-motion or spatial updating).

-The reason for depriving people of visual cues in some of the past experiments was: a) to prevent them from using landmarks rather than self-motion to remember target positions. b) to segregate the effect of vestibular senses in spatial updating from the other senses. We revised the sentence to state this more clearly. 

- Page 4, lines 2-4: can’t object distance change during rotations as well? I imagine, for example, rotating on a swivel chair in front of my desk: if I rotate by 180°, after rotation, I will not be able to grasp desk objects anymore, being them exactly behind me. Relative to the chair, the object probably maintained the position, but relative to my arm/hand, it actually changed, perhaps preventing me from grasping it. Am I missing what the authors are saying? Maybe it could be helpful to clearly define the “distance”.

-Here, we are considering the egocentric distance, i.e., distance from one’s self, to be the distance between the object and the observer’s head/body center. We revised the sentence to state this more clearly.

- A stronger and more thorough introduction to the reasons for conducting three different experiments and the use of both visual and physical motion information could help the reader to immediately grasp the take-home message of this work. In addition, it is not very clear which are the specific hypotheses and which is the novelty relative to the cited previous literature (references 16,17). Some new aspects are mentioned in the discussion (page 22, lines 4-6), but it would make more sense to insert them in the introduction session.

- We revised the introduction to provide more details in our reasonings and the hypothesis.

Method/Procedure:

- Page 9, line 4: can the authors clarify the reason for adding randomly generated dots on the screen? 

- Randomly generated dots were added to the screen to provide stronger visual cues (optic flow) during the lateral translation. We revised the text to clarify.

Was the screen the virtually rendered one?

- The screen was the virtually rendered one.

- What is the reason for changing the range of target displacements, increments, duration, and velocity of motion information across the experiments? 

- The range of target displacement was changed from 1m (Exp 1) to 0.47m (Exp 2) to compensate for the physical limitation of the moving platform.

- Why did the authors decide to make the dots blink in Experiment 3?

- The dots were made to blink in Exp 3 to prevent participants from using the dots as landmarks which may have otherwise helped localize the target’s positions after moving. We have revised the text to clarify.

- I do not think I got the virtual visual motion information: was it consistent with a real movement, or did it consist of moving the surroundings?

- The virtual visual motion information was consistent with the physical head translation experienced when the participants were physically moved on the motion platform in Exp 2. The visual motion was created by moving the virtual camera through the virtual surroundings for the visual movement only condition. We have revised the text to clarify this.

- How many trials for each condition in each experiment?

- Exp1: 112 trials, Exp2: 160 trials, and Exp3: 198 trials. We have added the number of trials to the text.

Data analysis:

- Page 13, line 10: Why did the authors consider errors deviating more than 2.5 SD only as a participant mistake and not as general variability?

- Some participants reported they had missed the target when it was presented and had responded randomly. Therefore, we considered errors that were too large (deviating more than 2.5 SD) as participant mistakes rather than general variability. 

- If the data from the two observer positions were brought to the same side, why one of the factors of the ANOVAs is the observer position? 

- During the experiment, the errors were recorded in left/right directions. Bringing them to the same side allowed us to make them dependent on the observer’s initial position, rather than dependent on the target position (left from the target vs. right from the target). The observer position factor in the ANOVA let us verify initial position did not impact these errors.

- Cannot be better to consider observer position levels like “same side vs different side” of the target instead of left and right?

- No, because the errors depend on where the target was relative to the observer. Even when the target was on the same side of space as the observer, the target could still be on the left or the right side of the observer (e.g., in Exp1 when the observer is at -0.5m, the target at -0.75m is on the left and the target at -0.25m is on the right side of the observer).

Results:

- Experiment 1: In Figure 3C, it seems that Visual Motion condition favor participants to reduce errors if compared with Stationary condition. If this is correct, the authors could mention this effect more explicitly, to clarify the direction of the reported difference between conditions.

- Although it the errors were reduced in the Visual Motion condition compared with the Stationary condition, we think this does not necessarily mean that they were more accurate. Participants made errors even in the Stationary condition, without moving at all. Since the targets were only shown at the beginning of each trial, they did not have a chance to correct any initial localization errors afterwards. We believe that the reduced errors (which were in the direction of the movement) were a consequence of an additional error (i.e., an updating error) resulting from being laterally translated. This is more clearly shown in Exp 3 where the errors in the Visual Motion conditions were again shifted in the direction of the movement but were increased relative to the Stationary condition.

- Figure 5A and B seem inverted: in Figure 5A, the participant seems to go from the center to the left; in Figure 5B from center to the right. If that’s correct, caption should be corrected.

- Corrected, thanks.

Discussion:

- I believe that a wider summary of the results is needed before the paragraph (page 21, lines 2-11) about Experiment 2 results.

- Added.

- Page 22, lines 15-17, “In Exp 1 these updating errors resulted in smaller errors for targets that became

less eccentric as a result of the movement, and larger errors for targets that became more

eccentric”: in Figure 3C, it seems the opposite. The red line for Visual Motion condition is closer to 0 as the position of the target is further from the initial position of the participant.

- In Fig. 3C, you can see that the mean error for the target at 1m (circled in blue) before moving (black dot) was about -0.22m. The mean error for the same target became closer to 0m after moving (red dot), i.e., smaller error. Before moving, the target at 1m was 1m away from the observer (from the Initial Position). After moving, the target was right in front of the observer (0m away from the Final Position), therefore becoming “less eccentric”. Hence, “smaller errors for targets that became less eccentric as a result of the movement.” You can see the similar pattern for targets at 1.25m and 0.75m, but opposite pattern for the targets at 0.25m, 0m, and -0.25m.

- Page 24, lines 3-6: The idea that potential physical motion could have played some role in spatial updating is not totally convincing to me. The actual physical movement did not improve participants’ performance (Exp 2), so why should the potential movement have a positive effect?

- Our results show smaller updating errors in Exp 2 (Visual Motion condition) compared to Exps 1 and 3. We believe this may be due to the potential movement enhancing vection [21] as shown in the past study. We have made this argument clearer in the discussion.

- The discussion section about the “updating ratio” (from page 25, line 7) seems to me more appropriate for a results section than the discussion.

- We have now moved it into the results section.

- Page 26, line 3: “1. 0”: typo

- Corrected.

- Page 29, line 10: the example should be about humans again, for better coherence.

- Revised.

- Generally, I would suggest to restructure the discussion to favor a better comprehension of the results as follows: Summary of general results � Results from Exp 1 and implications � Results from Exp 2 and implications � Results from Exp 3 and implications � general implications and interpretations. In the current discussion, the updating ratio computation and the first paragraph about Exp 2 come across as confusing.

- The discussion has been considerably revised.

Reviewer #2: This manuscript is an important advancement in the understanding of the mechanisms of spatial updating, especially as they relate to lateral translation movements. The authors conducted three experiments in a virtual reality setting, where they manipulated in various ways, the amount of (un)certainty about self-motion.

Major comments:

1. The introduction needs a bit more expansion, especially in terms what the previous studies that have investigated spatial updating with rotational movements and how the authors might envision spatial updating will change for lateral movements. This will help to motivate the study and the specific hypotheses more.

- Revised.

2. The authors indicate that they conducted three experiments and have one main hypothesis outlined in the introduction. The authors should tie each experiment to the hypothesis or expand and indicate why each of the three experiments was needed.

- Revised.

3. There are some details in the Methods section that need to be added (e.g., the number of trials in total; whether the conditions blocked or mixed; the number of repetitions per trial type; the rooms that the experiments took place in; etc.), as well as motivation behind their approaches (and changes across the experiments). For example, why were the distances between the targets chosen, and changed, in the three experiments? How do each of the changes contribute to answering how spatial updating for lateral movements? These specific methodological motivations are important for understanding how each experiment is motivated and in what way it contributes to answering the overarching question of the manuscript.

- We revised the Methods section to add more detail.

4. For experiment 2, could the authors indicate the specific progression of information for the different conditions tested and the effect on spatial updating. For example, was the physical and visual motion condition expected to reduce uncertainty around self-motion the most and therefore, produce the smallest errors; if so, was the next best condition considered to be the physical only condition, etc.? How do the results compare to these expectations? These are very briefly covered in the discussion.

- Typically, self-motion is overestimated with only visual or physical motion cues (larger overestimation with only physical motion cues) compared to when moving with both cues available. We expected smallest error with full motion (physical and visual motion condition), followed by vision only condition, then largest error with physical only condition. This has now been added in the manuscript.

Minor point:

1. Could the authors provide an explanation as to why the experiments were conducted in virtual reality settings and not in the actual environments?

- We conducted the experiment in VR to make it possible to provide visual motion (optic flow) without physical motion which is difficult to do in the actual environment (e.g., moving an entire room while the observer is sitting still). 

2. Why were controllers used in the first two experiments, but not in the third? How does this help to disentangle the spatial updating process for lateral movements?

- Controllers were used in Exps 1 and 2 because it was an easy and accurate method for participants to indicate the remembered position of the targets. However, we had complaints from some participants that their arms and shoulders became tired by the end of the experiment which took about an hour. This became a concern as the number of trials increased in the later experiments. Before Exp 3, we conducted a short pilot study to see whether the performance differed between the two methods (controller vs. keyboard) and did not find significant difference. Therefore, in Exp 3 we decided to use keyboard instead of controllers for participant responses.

Reviewer #3: Kim and Harris report that during lateral translation, subjects systematically underestimate target eccentricity, particularly for more eccentric targets, with errors influenced by whether the target crosses the observer's midline. These errors likely reflect changes in spatial updating tied to perceived self-motion and reconstructing target positions across visual fields. While the results are interesting, the manuscript would benefit from clearer data representation, both in the text and figures. Additionally, the writing (several errors) could be substantially improved (could be more compact) for better clarity and impact.

Major issues:

EXPERIMENT 1:

• The text is not clear. What is the spacing between targets? I agree that it is mentioned in the “methods’ but it is good to remind the reader of the relationships between target positions, travel range, and motion simulation. 

- Revised.

My major concern here is the overinterpretation of the main effect (page 15, lines 15-16) as reported by a significant interaction (motion and target) with p = 0.045. I suggest emphasizing the small effect size in the interaction, and a much clearer explanation for spatial updating. 

- Our interpretation of the updating being dependent on the travel distance and target eccentricity was not just based on the significant interaction between motion and target. We followed it up with post-hoc evaluations which revealed more complex relationships between the two factors (2 motion conditions X 7 target positions, resulting in 86 total comparisons). Some more results from post-hoc analysis have been added to the results section to make this clearer.

What are the data points to the extremes of these lines (- 0.75 and 0.75), though the translation is between – 0.5 and 0.5? 

- Having targets beyond as well as within the translation range allowed us to measure updating for targets that cross or not cross the hemi-field. This is shown in Fig 10.

Since all the subjects are right-handed, was there any hemi-field effect (it is reported that errors increased with eccentricity, it seems there is no hemifield effect but could be mentioned / investigated further)? 

- There was no difference in errors between the targets in the left or the right field from the observer. We interpreted th

---

## [Decision Letter · Decision Letter 1]

7 Nov 2024

PONE-D-24-28073R1Updating the remembered position of targets following passive lateral translationPLOS ONE

Dear Dr. Kim,

Thank you for submitting your manuscript to PLOS ONE. After careful consideration, we feel that it has merit but does not fully meet PLOS ONE’s publication criteria as it currently stands. Therefore, we invite you to submit a revised version of the manuscript that addresses the minor points raised during the review process. Please refer below to some minor issues that are spotted by Reviewer 2 and prepare a revised version of the manuscript based on these remarks. 

We look forward to receiving your revised manuscript.

Kind regards,

Dimitris Voudouris

Academic Editor

PLOS ONE

Journal Requirements:

Reviewers' comments:

Reviewer's Responses to Questions

**Comments to the Author**

1. If the authors have adequately addressed your comments raised in a previous round of review and you feel that this manuscript is now acceptable for publication, you may indicate that here to bypass the “Comments to the Author” section, enter your conflict of interest statement in the “Confidential to Editor” section, and submit your "Accept" recommendation.

Reviewer #2: (No Response)

Reviewer #3: All comments have been addressed

2. Is the manuscript technically sound, and do the data support the conclusions?

Reviewer #2: Yes

Reviewer #3: Yes

3. Has the statistical analysis been performed appropriately and rigorously? 

Reviewer #2: Yes

Reviewer #3: Yes

4. Have the authors made all data underlying the findings in their manuscript fully available?

Reviewer #2: Yes

Reviewer #3: No

5. Is the manuscript presented in an intelligible fashion and written in standard English?

Reviewer #2: Yes

Reviewer #3: Yes

6. Review Comments to the Author

Reviewer #2: The authors have made great efforts to address the comments.

I have a few minor points left:

1. The section on the updating ratio should be moved to the analysis (Methods) section - it sticks out in its current position.

2. While I appreciate that the authors are trying to highlight the similarities and differences in the procedure section for the three experiments, I would prefer a more detailed description for each experiment first, followed by a short section where similarities and differences are highlighted. This will make the interpretation of the results even easier.

3. A summary paragraph at the end of the discussion section would be useful.

4. Some additional proofreading is required; though, these are very minor.

Reviewer #3: The authors have addressed all my comments and concerns and also provided further clarifications. This work significantly enhances our understanding of perceptual processes during self-motion especially during lateral translation. Good work.

7. PLOS authors have the option to publish the peer review history of their article (what does this mean?). If published, this will include your full peer review and any attached files.

Reviewer #2: No

Reviewer #3: No

---

## [Author Response · Author response to Decision Letter 1]

23 Nov 2024

Response to Reviewers:

Reviewer #2: The authors have made great efforts to address the comments.

I have a few minor points left:

1. The section on the updating ratio should be moved to the analysis (Methods) section - it sticks out in its current position.

- Revised.

2. While I appreciate that the authors are trying to highlight the similarities and differences in the procedure section for the three experiments, I would prefer a more detailed description for each experiment first, followed by a short section where similarities and differences are highlighted. This will make the interpretation of the results even easier.

- The procedure section now has 3 subsections to explain each experiment separately.

3. A summary paragraph at the end of the discussion section would be useful.

- We added a conclusion section after the discussion that summarize the main findings.

4. Some additional proofreading is required; though, these are very minor.

- Revised.

Reviewer #3: The authors have addressed all my comments and concerns and also provided further clarifications. This work significantly enhances our understanding of perceptual processes during self-motion especially during lateral translation. Good work.

- N/A

---

## [Decision Letter · Decision Letter 2]

12 Dec 2024

Updating the remembered position of targets following passive lateral translation

PONE-D-24-28073R2

Dear Dr. Kim,

We’re pleased to inform you that your manuscript has been judged scientifically suitable for publication and will be formally accepted for publication once it meets all outstanding technical requirements.

Kind regards,

Dimitris Voudouris

Academic Editor

PLOS ONE

Additional Editor Comments (optional):

Reviewers' comments:

Reviewer's Responses to Questions

**Comments to the Author**

1. If the authors have adequately addressed your comments raised in a previous round of review and you feel that this manuscript is now acceptable for publication, you may indicate that here to bypass the “Comments to the Author” section, enter your conflict of interest statement in the “Confidential to Editor” section, and submit your "Accept" recommendation.

Reviewer #2: All comments have been addressed

2. Is the manuscript technically sound, and do the data support the conclusions?

Reviewer #2: Yes

3. Has the statistical analysis been performed appropriately and rigorously? 

Reviewer #2: Yes

4. Have the authors made all data underlying the findings in their manuscript fully available?

Reviewer #2: Yes

5. Is the manuscript presented in an intelligible fashion and written in standard English?

Reviewer #2: Yes

6. Review Comments to the Author

Reviewer #2: (No Response)

7. PLOS authors have the option to publish the peer review history of their article (what does this mean?). If published, this will include your full peer review and any attached files.

Reviewer #2: No

---

## [Editor Report · Acceptance letter]

16 Dec 2024

PONE-D-24-28073R2 

PLOS ONE

Dear Dr. Kim, 

I'm pleased to inform you that your manuscript has been deemed suitable for publication in PLOS ONE. Congratulations! Your manuscript is now being handed over to our production team.

Kind regards, 

on behalf of

Dr. Dimitris Voudouris 

Academic Editor

PLOS ONE